



# An improved subgrid channel model with upwind form artificial diffusion for river hydrodynamics and floodplain inundation simulation

Youtong Rong, Paul Bates, Jeffrey Neal

School of Geographical Sciences, University of Bristol, Bristol, BS8 1QU, UK

*Correspondence to*: Youtong Rong (youtong.rong@bristol.ac.uk)

**Abstract.** An accurate estimation of river channel conveyance capacity and the water exchange at the river-floodplain interfaces is pivotal for flood modelling. However, in large-scale models limited grid resolution often means that small-scale river channel features cannot be well represented in traditional 1D/2D schemes. As a result instability over river and floodplain

boundaries can occur, and flow connectivity, which has a strong control on the floodplain hydraulics, is not well-approximated. A subgrid channel model (SGC) based on the local inertial form of the shallow water equations, which allows utilization of approximated sub-grid scale bathymetric information while performing very efficient computations has been proposed as a solution, and it has been widely applied to calculate the wetting and drying dynamics in river-floodplain systems at regional scales. Unfortunately, SGC approaches to date have not included latest developments in numerical solutions of the local inertial

equations, and the original solution scheme was reported to suffer from numerical instability in low friction regions such as urban areas. In this paper, for the first time, we implement a newly developed diffusion and explicit adaptive weighting factor in the SGC model. An adaptive artificial diffusion is explicitly included in the form of an upwind solution scheme based on the local flow status to improve the numerical flux estimation. A structured sequence of numerical experiments is performed, and the results confirm that the new SGC model improved the model performance in terms of water level and inundation

extent, especially in urban areas where the Manning parameter is less than $0.03\mathrm{m}^{-1/3}\mathrm{s}$. By not compromising computational efficiency, this improved SGC model is a compelling alternative for river-floodplain modelling, particularly in large-scale applications.

## 1 Introduction

Recent flood events and climate change concerns have boosted the requirements for hydraulic models with fast and accurate

calculation of both spatial and temporal flow dynamics in river-floodplain systems (Jongman, Ward and Aerts 2012, Edmonds et al. 2020, McMichael et al. 2020). Flood risk assessment based on the output of hydrodynamic simulation models has been proven effective to prepare for disasters and can facilitate good decision making at local, regional, and national levels of government, reducing the risk posed by flood hazards (Al Baky, Islam and Paul 2020). There is thus an increasing demand for flood modelling studies that can accurately represent the dominant hydrodynamic process during flood events and provide



recommendations for mitigating measures to alleviate the impact of potential flooding (Paiva, Collischonn and Tucci 2011, Yamazaki et al. 2014).

Though the large magnitude associated with most floods might appear to overshadow the impact of river channel bathymetry incorporation, the channel in fact still conveys a significant proportion of the flow during a flood event because of the much higher channel velocity compared to the floodplain (Grimaldi et al. 2018, Neal et al. 2015). As a result, accurate estimates of

river conveyance capacity and cross-section depth values deserve more attention than the accurate predictions of far-field flood elevations, and physics-based hydrodynamic model performance can be improved in terms of wave propagation speed and inundation extent through good representation of the river channel (Fewtrell et al. 2011). For example, the average predicted inundation extent decreased by more than 30% and average water surface elevation dropped by 0.5 m after incorporating bathymetric data in a hydrodynamic model of Strouds Creek in North Carolina (Cook and Merwade 2009). Changes in

inundation extent due to proper accounting of river channel conveyance are much greater for areas with a flat topography. Crucial physical aspects of river hydraulics, like backwater effects and looped stage-discharge relations, are omitted without the inclusion of the river bathymetry or even with simplified river routing models (e.g., Muskingum-Cunge method, kinematic hydraulic model). Such a simplified hydrodynamic model cannot therefore resolve inundation patterns necessary to understand associated risks locally (Schumann et al. 2013). Furthermore, floodplain water levels cannot be assumed to be the same as

channel water levels because of water storage on floodplains, and complex and nonlinear interactions between the channel and floodplain are to be expected (Alsdorf et al. 2007, Trigg et al. 2009, Trigg et al. 2012). Efficient incorporation of river channel bathymetry in flood inundation models is therefore fundamental for modelling the mass and momentum exchange at the river-floodplain interface, dynamic wetting and drying process on the floodplain and the wave propagation characteristics.

Given that river channel flows are an essential component in flood modelling, several approaches have been implemented to integrate river hydraulics into hydrodynamic models:

(1) a 1D river hydraulic model without floodplain or with extended cross sections to approximate floodplain storage and river channel conveyance in 1D

(2) a 2D floodplain inundation model representing channel and floodplain in a single discretization

(3) 1D/2D representation with main channels and floodplain

(4) 1D/2D representation with sub-grid scale simplified channels

By omitting floodplain inundation process, 1D river hydraulic models are a lightweight alternative to a 2D hydrodynamic model framework. Together with a volume storage grid or an extended cross section to approximate the floodplain storage and conveyance, the 1D model has been successfully implemented to assess flood risk in major rivers globally (Yamazaki et al.

2011, Roberto Rudari et al. 2015). However, 1D models are incapable of accommodating the real physical and hydrodynamic conditions required to represent a number of river processes (Merwade, Cook and Coonrod 2008). Realistic flow inundation process and channel-floodplain momentum exchange cannot be obtained without the floodplain component. With the emphasis on the floodplain inundation process, 2D models provide a solution which can either ignore the river conveyance capacity or




represent the channel with a fine grid resolution at the cost of substantially increasing computational time, even with an
unstructured discretization of space. This costly and unnecessary grid refinement in the channel region has hampered the
further application of full 2D models, especially for resource-intensive large-scale flood inundation simulation.

The combination of a 1D model in the channel and a 2D model for the floodplain offers the benefits of capturing 2D processing
on the floodplain whilst minimising the computational costs and below water line data requirements in the river channel.
However, the troublesome interactions at the river-floodplain interface demand extra attention due to their effect on the mass
and momentum balance there, otherwise a divergent outcome can be easily acquired. Furthermore, 1D/2D models can have
limited capacity to properly represent minor river channels that can have a strong control on the floodplain hydraulics and this
restricts the further improvement of 1D/2D model efficiency. Since models missing either the channel network or floodplain
component have reduced predictive skill at large-scales (Neal, Schumann and Bates 2012), research has sought to identify an
efficient alternative for river-floodplain modelling.

To enable a physically-consistent representation of the river-floodplain system and reduce the computational burden of
floodplain inundation modelling, the subgrid channel (SGC) model solving the local inertial form of the shallow water
equations has been proposed, allowing the utilization of available sub-scale bathymetric information while performing
computations on relatively coarse grids (Neal, Schumann and Bates 2012). Flow connectivity provided by the fine resolution
river channel network and its strong control on the floodplain hydraulics is incorporated into the model. Precise mass balance
in regions where wetting and drying occurs is achieved within a well-structured, mildly nonlinear system for the discrete water
surface elevation. The adoption of the subgrid method improves computing performance by roughly a factor of 20 compared
with the classical 2D model based on unstructured grids (Sehili, Lang and Lippert 2014).

Unfortunately, SGC approaches to date have not included the latest developments in numerical solutions of the local inertial
equations. Despite its high performance, the original SGC solution scheme was reported to suffer from numerical instability
in low friction regions such as the urban areas (Bates, Horritt and Fewtrell 2010). The problem has been tackled by introducing
limited artificial diffusion in the form of weighting factors for the neighbouring flux(de Almeida et al. 2012). An explicit
expression of the adaptive weighting factor depending on local velocity, flow depth, grid resolution, and time step size, has
also recently been developed to recognize the different diffusion needed at each iteration (Sridharan et al. 2020). This adaptive
weighting factor has been shown to improve simulation of flood propagation over the floodplain, but its application to the
channel hydrodynamics calculation is still lacking, as well as its ability to assess the water exchange between river and
floodplain. Implementation of the schemes above also has not yet been evaluated at channel confluences. This paper therefore
adds this new explicitly calculated diffusion to an SGC model for the first time, and also adds a further constraint on the
available range of the weighting factor to balance the contribution of the flux from the local and upwind surfaces. This enables
an improved and computationally efficient solution for river-floodplain flow inundation simulation with multi-core CPUs.

The paper is structured as follows: the development of the improved SGC model is outlined in section 2, with the emphasis on
how to adopt the new upwind solution scheme in the discretization procedure of the governing equations. A wide range of
tests are set up in section 3, from steady/unsteady problems with an analytical solution to practical applications with detailed


ground survey data, to evaluate the model performance quantitively. Volume error, Root Mean Square Error(RMSE) and inundation extent are used to quantify the model accuracy compared with three other solvers which are also implemented in
the LISFLOOD-FP hydrodynamic model. Conclusions are drawn in section 4.

**2 Methodology**

The following methodology section outlines the steps associated with including the latest development in numerical solutions of the local inertial equations into the SGC model. Procedures adopted to improve the original SGC solution schemes in the calculation of floodplain flow propagation, river hydrodynamics, and water exchange over the river-floodplain boundaries are
described respectively. These approaches have been confirmed to improve the accuracy and robustness of the solution scheme and are therefore here implemented to enhance the performance of the SGC model.

**2.1 Governing equations**

The subgrid channel model employs the efficient local inertial formulation of the shallow water equations to calculate the surface flux between adjacent floodplain cells and update the water depth in every structured DEM cell, as shown in equations
(1)-(3). The simplified governing equations are achieved by neglecting the advection term in the momentum equation from the quasi-linearized 1D Saint-Venant equations (Bates et al. 2010). For gradually varied flows where such approximations are not contradicted (De Almeida and Bates 2013), these equations can efficiently yield results with accuracy comparable to the full-dynamic system (Neal et al. 2012, Rajib et al. 2020). The equations with formulae decoupled in the x and y directions can be employed directly for the calculation of flood propagation over the floodplain. A 1D interpretation of these governing
equations is required for the river flow calculation, and the flow area is included explicitly in the solution scheme during the discretization process to account for precise channel conveyance capacity.

$$\frac{\partial h}{\partial t} + \frac{\partial q_x}{\partial x} + \frac{\partial q_y}{\partial y} = 0, \tag{1}$$

$$\frac{\partial q_x}{\partial t} + gh\frac{\partial (h+z)}{\partial x} + \frac{gn^2|q_x|q_x}{h^{7/3}} = 0, \tag{2}$$

$$\frac{\partial q_y}{\partial t} + gh\frac{\partial (h+z)}{\partial y} + \frac{gn^2|q_y|q_y}{h^{7/3}} = 0, \tag{3}$$

Where $h$ is the water depth [L], $q$ is the discharge per unit width [$L^2T^{-1}$], $z$ is the bed elevation [L], $g$ is the acceleration due to gravity [$LT^{-2}$], $n$ is the Manning friction coefficient [$L^{-1/3}T$], $x/y$ denotes the horizontal/vertical coordinate [L], and $t$ is the time [T].

Flood inundation models governed by local inertial equations focus on the calculation of the dominant hydrodynamics process during gradually varied and sub-critical flow propagation. Accurate mass and momentum balance in shallow flows over
complex geometries is assured in the presence of wetting and drying. Compared with the full-dynamics SWEs, the simplified governing equations demand less computational resources while still preserving the main features of the gradually varied flow field (Shaw et al. 2021). Therefore, the local inertial equations are taken as the governing equations for the calculation of flow





propagation in the new SGC model. Latest developments in solution schemes for the local inertial equations are implemented to tackle the potential instability problem of the original SGC model of Bates et al. (2010).

### 2.2 Solution scheme for floodplain inundation calculation

The original SGC solution scheme is reported to suffer from divergence problems in urban areas with low friction, and simulation accuracy is also impacted by the original solution scheme. Improvements to the original SGC solution schemes (equations (2) and (3)) follow the procedure proposed by de Almeida et al. (2012), which explicitly includes artificial diffusion in the form of an upwind solution scheme. A fixed weighting factor balancing the contribution of upwind and local flow flux is necessary in this solution scheme, and repeated tests are required to determine a global optimum value for the weighting factor. Then an adaptive weighting factor depending on the local flow status is explicitly integrated to enable an automatic determination of the artificial diffusion needed to stabilize the solution scheme (Sridharan et al. 2020). By importing artificial diffusion, the explicit expression of the momentum equations in the form of the upwind scheme is acquired as shown in equation (4). A similar equation can be derived with the same structure to calculate the discharge in the y-direction. With the upwind discharge included ($Q_{i-3/2,j}^t$ or $Q_{i+1/2,j}^t$, depending on local flow direction), the system can respond to the upwind flow flux variation with much flexibility to avoid the formulation of non-physical water depth gradients caused by the delayed propagation of flow information in the original scheme. Imported adaptive diffusion enables oscillation-free solutions in many cases where the original scheme has the potential to be divergent (O'Loughlin et al. 2020, Shustikova et al. 2019, Sridharan et al. 2021).

$$Q_{i-1/2,j}^{t+\Delta t} = \begin{cases} \dfrac{\theta Q_{i-1/2,j}^t + (1-\theta)Q_{i-3/2,j}^t - gA_{i-1/2,j,flow}^t \Delta t S_{i-1/2,j}^t}{\left\{1+g\Delta t n_{i-1/2,j}^2 \left|Q_{i-1/2,j}^t\right|/\left[(h_{i-1/2,j,flow}^t)^{4/3}A_{i-1/2,j,flow}^t\right]\right\}}, if\ Q_{i-1/2,j}^t > 0, \\[3mm] \dfrac{\theta Q_{i-1/2,j}^t + (1-\theta)Q_{i+1/2,j}^t - gA_{i-1/2,j,flow}^t \Delta t S_{i-1/2,j}^t}{\left\{1+g\Delta t n_{i-1/2,j}^2 \left|Q_{i-1/2,j}^t\right|/\left[(h_{i-1/2,j,flow}^t)^{4/3}A_{i-1/2,j,flow}^t\right]\right\}}, if\ Q_{i-1/2,j}^t < 0, \\[3mm] \dfrac{-gA_{i-1/2,j,flow}^t \Delta t S_{i-1/2,j}^t}{\left\{1+g\Delta t n_{i-1/2,j}^2 \left|Q_{i-1/2,j}^t\right|/\left[(h_{i-1/2,j,flow}^t)^{4/3}A_{i-1/2,j,flow}^t\right]\right\}}, if\ Q_{i-1/2,j}^t = 0 \end{cases} \quad (4)$$

Here, $\Delta x$ is the horizontal dimensions of the DEM cells [L], and $S$ is the water surface slope between two adjacent cells. $A$ is the flow area [L$^2$], and $h$ is the water depth [L]. The subscript of $flow$ indicates these components varies with the flow status. $Q$ is discharge across the cell boundaries [L$^3$T$^{-1}$]. Depending on the flow direction at the local cell interface, one of its two adjacent surfaces in the same direction is selected as the upwind flux, representing the information propagation direction of the local flow field. No artificial diffusion is included if there is no flux at the local cell interface. With the momentum equations solved based on the discharge from the last time step, local discharge representing the momentum exchange at the grid interfaces is updated, and then the cell-average water depth can be solved with equation (5), which is the continuity equation relating flow into a cell and its volume change.

$$h_{i,j}^{t+\Delta t} = h_{i,j}^t + \Delta t \frac{Q_{x\ i-1/2,j}^{t+\Delta t} - Q_{x\ i+1/2,j}^{t+\Delta t} + Q_{y\ i,j-1/2}^{t+\Delta t} - Q_{y\ i,j+1/2}^{t+\Delta t}}{A_{i,j}}, \quad (5)$$



Following Sridharan et al. (2020), the weighting factor $\theta$ defined in equation (6) quantifies how much artificial diffusion is required to stabilize the solution scheme. A predefined uniform weighting factor was previously set for surface flux calculation in the original subgrid channel scheme of the LISFLOOD-FP model, and a tricky calibration was required to acquire a global optimal solution (de Almeida et al. 2012). However a global optimum solution does not guarantee the best performance of the local flow calculation so an adaptive procedure is now implemented to determine the value of $\theta$ so that no trials and

approximations are needed and the weighting factor can be updated automatically based on the local velocity, flow depth, grid resolution, and time step size (Sridharan et al. 2020), shown in equation (6). An extra constraint on the feasible range of the weighting factor $\theta$ is applied in this paper to limit its minimum value to 0.7. This ensures that the local interface flux dominates the flux calculation while artificial diffusion from upwind cannot be overused. Potential instability problems can be induced on the condition that the local flux estimation mainly depends on the upwind flow information while ignoring the local flow

status. These adaptive measures pay close attention to the change of the flow field to identify the dominant factor for updating the surface flux, thus increasing the robustness of the system. Compared with the fixed weighting factor strategy, much flexibility is incorporated without compromising the computational efficiency.

$$\theta_{i-1/2,j} = 1 - \frac{\Delta t}{\Delta x} \min\left(\frac{|q_{i-1/2,j}|}{h_{flow}}, \sqrt{gh_{flow}}\right), \tag{6}$$

### 2.3 Solution scheme for river hydraulics calculation

Similar procedures implemented to improve floodplain inundation calculation are applied to discretize the 1D interpolation of the local inertial equations. Sub-scale channel parameters that represent the rectangular channel flow area ($A$) are integrated during the discretization process, accounting for the flow conveyance capacity with sub-scale river width ($w$) and flow depth ($d$). The sub-scale representation of the channel features enables a numerically consistent representation of flow dynamics even in large-scale river-floodplain modelling. With sub-grid sampling, the underlying topography dominating the river flow

transport, in the form of the approximated rectangular river channel, is still utilized despite a coarser grid resolution being applied for the floodplain. This allows the user to focus details where required for inundation extent prediction without compromising computational efficiency in locations of little topographic variability. A more stable and efficient solution scheme is acquired by this means for the river hydraulics calculation (equation (7)).

$$Q_{c\,i,j-1/2}^{t+\Delta t} = \begin{cases} \dfrac{\theta Q_{c\,i,j-1/2}^{t} + (1-\theta)Q_{c\,up}^{t} - gA_{c\,i,j-1/2,flow}^{t}\Delta t S_{c\,i,j-1/2}^{t}}{\left\{1 + g\Delta t n_{c\,i,j-1/2}^{2}\left|Q_{c\,i,j-1/2}^{t}\right| / \left[(R_{c\,i,j-1/2,flow}^{t})^{4/3}A_{c\,i,j-1/2,flow}^{t}\right]\right\}}, & if\ \left|Q_{c\,i,j-1/2}^{t}\right| > 0; \\[4mm] \dfrac{-gA_{c\,i,j-1/2,flow}^{t}\Delta t S_{c\,i,j-1/2}^{t}}{\left\{1 + g\Delta t n_{c\,i,j-1/2}^{2}\left|Q_{c\,i,j-1/2}^{t}\right| / \left[(R_{c\,i,j-1/2,flow}^{t})^{4/3}A_{c\,i,j-1/2,flow}^{t}\right]\right\}}, & if\ Q_{c\,i,j-1/2}^{t} = 0, \end{cases} \tag{7}$$

Here, $i$ and $j$ denote the column and row index of the gird respectively, which is the same as the 2D base model. The subscript $c$ denotes the channel flow discharge, to distinguish it from the floodplain surface flux. Only if two adjacent cells are river cells would the equation (7) be employed to calculate the river channel flux. $A$ is the channel flow area [$L^2$], and $R$ represents the hydraulic radius [$L$]. $Q_{c\,up}$ is the upwind flow discharge [$L^3T^{-1}$]. Quite different from the floodplain flux calculation where





the whole cell is used to transfer the flux, the river channel discharge only occupies a proportion of a grid, depending on the

ratio of the channel width to the cell dimension. The channel flow area quantifies the flow conveyance capacity as the product

of the river width and flow depth. Overbank flow happens if the channel flow depth exceeds the bank elevation. Due to the

deficiencies of typical bathymetry data and difficulties in field survey to acquire continuous bathymetric information for flood

inundation simulation, a method is provided for the estimation of the channel depth. By approximating the river channel as a

rectangular geometry, the channel width and channel bed elevation can be estimated as a function of the upstream discharge

using empirical relations or gradually varied flow theory (Leopold and Maddock 1953, Neal et al. 2021). Approximated

channel features provide a feasible solution for simulating river hydraulics over regional-to-continental scale domains and in

data-sparse areas. Externally specified bathymetry either from field survey or some form of estimation process are also

applicable if available.

Special attention should be paid to determining the upwind surface flux in the river channel. As depicted in Figure 1, the cells

with the green colour represent the subgrid cells where the channel width is narrower than the grid resolution. For simplicity

a uniform river width is applied for sub-grid cells. The discharge is defined as positive if the flow direction is from west to

east or from north to south. To calculate the discharge $Q_{c\,i,j-1/2}$, a combination of the surface discharge $Q_{c\,i-1/2,j-1}$, $Q_{c\,i,j-3/2}$ ,

$Q_{c\,i+1/2,j-1}$ is utilized if $Q_{c\,i,j-1/2} > 0$ which is flowing from north to south for the local cell surface flux (equation (8)). While

if the discharge $Q_{c\,i,j-1/2} < 0$, the flow propagation information comes from $Q_{c\,i-1/2,j}$, $Q_{c\,i,j+1/2}$, $Q_{c\,i+1/2,j}$, which is the

opposite flow propagation direction (equation (9)). Hence, the upwind flow discharge is determined by judging the sign of the

local cell surface discharge. Quite different from the 2D base model which is decoupled in the x and y direction and upwind

flow information comes from either the horizontal or longitudinal direction, the river channel should combine all the possible

upwind flow discharge, though only one upwind flow discharge is in effect (the interface flux is not zero) for a single channel

without a river confluence, as is shown in Figure 1(b). At river confluences, discharge variation in three upwind surfaces are

responsible for mass balance there, and all the possible sources of upwind surface flux should be considered (Figure 1(a)).



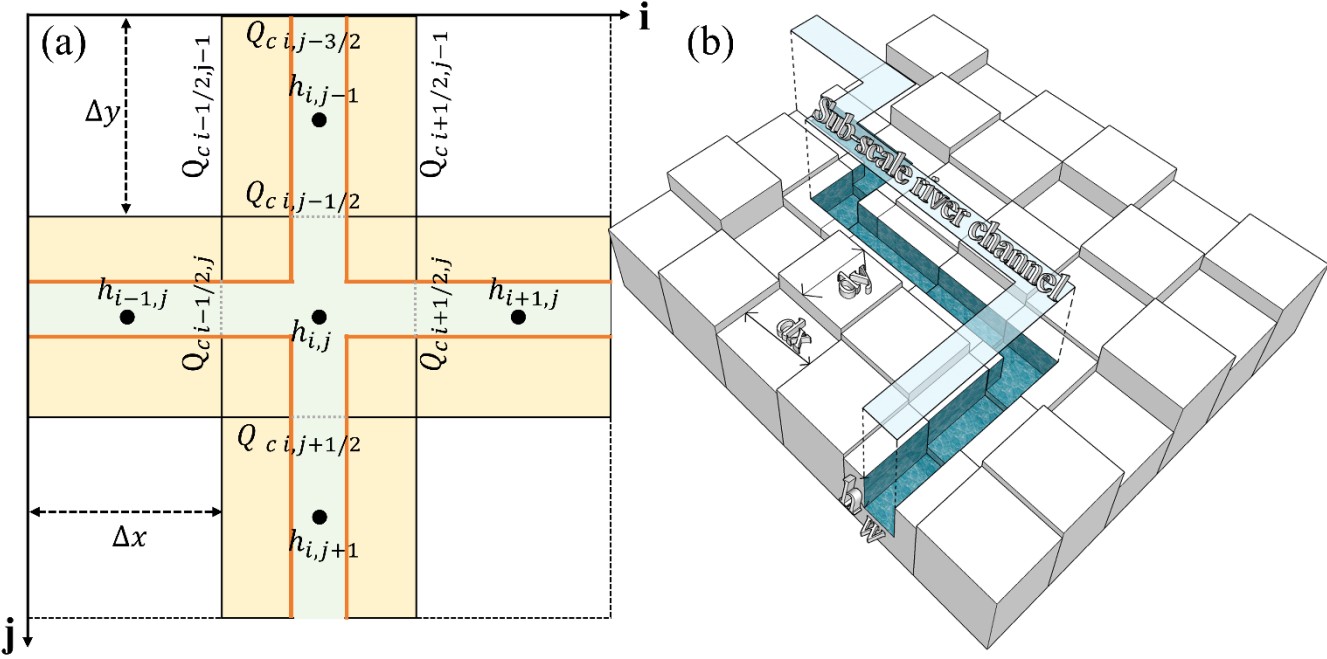

**Figure 1: Sub-scale representation of the river channel on a coarse resolution DEM. Sub-grid cells are in shaded colour, with the black line showing the boundary of the DEM cells. Green shadow represents the river channel, while the yellow shadow represents the floodplain proportion with a sub-grid cell. (a) a channel network with the river confluence. (b) a perspective view of a sub-scale river channel on the DEM grid.**

Depending on the sign of the local interface flux, $Q_{c\,up}$ which represents the source where the flow information comes from is selected. The sign of each item in equations (8) and (9) is dependent on the positive definition of flow direction. In fact, $Q_{c\,i-1/2,j-1}$ and $Q_{c\,i+1/2,j-1}$ in equation (8) is zero as no river discharge exchange exists through these two interfaces. An extra minus sign is added to the right-hand side of equation (9) to coincide with local interface flux where $Q_{c\,i,j-1/2} < 0$, to ensure that the upwind flow information is not contradicted with the local flow status.

$$Q_{c\,up} = Q_{c\,i-1/2,j-1} + Q_{c\,i,j-3/2} - Q_{c\,i+1/2,j-1},\ if\ Q_{c\,i,j-1/2} > 0, \tag{8}$$

$$Q_{c\,up} = -\left(Q_{c\,i-1/2,j} - Q_{c\,i,j+1/2} - Q_{c\,i+1/2,j}\right),\ if\ Q_{c\,i,j-1/2} < 0, \tag{9}$$

It is also noteworthy that upwind flow may still have a different direction to the local flow following equations (8) and (9). For example, all the four interface fluxes leave the current cell if there exists a point source inside the cell, and the flow propagation direction is from the current cell to its four neighbour cells. Under such a situation it is apparent that all the interface fluxes have a unique flow direction, and no upwind flow propagation information is available for any of these four interfaces. Hence, further confinement is implemented to avoid the abuse of the contradicted upwind flow information (equation (10)), or the upwind discharge would induce an unsteady flow condition, making the solution divergent. This constraint also applies to the





floodplain momentum flux calculation. In fact, the numerical solution returns to the scheme employed in the original SGC model under such circumstances.

$$Q_{c\,up} = 0 \; and \; \theta = 1.0, \; if \; Q_{c\,i,j-1/2} * Q_{c\,up} < 0, \tag{10}$$

Following the aforementioned procedures, wave propagation in the river and flood inundation over the floodplain can be calculated independently. No water exchange between the river channel and the floodplain exists at this stage. The interaction of flow information between the river and floodplain is accomplished inside the subgrid cells, as shown in the following section.

## 2.4 Water exchange at the river-floodplain interface

Subgrid cells can be used to route the flood wave downstream with the river channel bathymetry, and the remaining proportion of the cell can be used for floodplain flow propagation once the bankfull depth is reached, as shown in Figure 1. The floodplain proportion in a sub-grid cell is utilized to conduct the momentum exchange with its neighbouring floodplain cells, while the river channel part connects the upstream and downstream river cells. As the floodplain components only take up a proportion of the subgrid cells, the capacity to transfer the momentum is to some extent reduced, depending on the proportion of the remaining width $(\Delta x - w_e)$. The river channel and the floodplain component in a subgrid cell are working together to update the water depth, and same water surface elevation is shared by these two components when overbank flow happens. Therefore, the water exchange is fulfilled implicitly without calculating the mass balance at the coupling interface to redistribute the water volume. Compared with a tricky treatment in the river-floodplain interface, the water exchange in the SGC model applies a simple storage cell that distributes the water volume explicitly. Momentum loss is neglected at the river-floodplain interface and this is a reasonable critique of the scheme but implemented for simplicity.

$$Q_{i-1/2,j}^{t+\Delta t} = \begin{cases} \dfrac{\theta Q_{i-1/2,j}^{t} + (1-\theta)Q_{i-3/2,j}^{t} - g\Delta t S_{i-1/2,j}^{t} h_{flow}^{t}(\Delta x - w_e)}{\left\{1 + g\Delta t n_{i-1/2,j}^2 \left|Q_{i-1/2,j}^t\right| / \left[(h_{i-1/2,j,flow}^t)^{4/3} A_{i-1/2,j,flow}^t\right]\right\}}, & if \; Q_{i-1/2,j}^t > 0, \\[3ex] \dfrac{\theta Q_{i-1/2,j}^{t} + (1-\theta)Q_{i+1/2,j}^{t} - g\Delta t S_{i-1/2,j}^{t} h_{flow}^{t}(\Delta x - w_e)}{\left\{1 + g\Delta t n_{i-1/2,j}^2 \left|Q_{i-1/2,j}^t\right| / \left[(h_{i-1/2,j,flow}^t)^{4/3} A_{i-1/2,j,flow}^t\right]\right\}}, & if \; Q_{i-1/2,j}^t < 0, \\[3ex] \dfrac{-g\Delta t S_{i-1/2,j}^{t} h_{flow}^{t}(\Delta x - w_e)}{\left\{1 + g\Delta t n_{i-1/2,j}^2 \left|Q_{i-1/2,j}^t\right| / \left[(h_{i-1/2,j,flow}^t)^{4/3} A_{i-1/2,j,flow}^t\right]\right\}}, & if \; Q_{i-1/2,j}^t = 0 \end{cases} \tag{11}$$

Here $w_e$ is the smaller channel width between two neighbor subgrids [L]. $(\Delta x - w_e)$ represents the remaining width perpendicular to the flow direction which can be used to transfer the floodplain momentum. With the same governing equation for the river hydraulics and floodplain inundation calculations, the computational timestep in each component matches with each other, thereby increasing the robustness of the model. In every iteration, the coordinated timestep is controlled by the CFL condition to ensure the stability of the numerical scheme. The value of $cfl$ in equation (12) is set to 0.7 for the simulations reported in this, which is a trade-off between computational efficiency and numerical stability, but can be adjusted in the model by the user.





$$\Delta t = cfl \, \frac{\Delta x}{\sqrt{gh_{max}}}, \tag{12}$$

At every iteration, the time step is configured according to the maximum water depth over the domain. A predefined timestep is utilized for the initial dry bed domain. At every time step, following the procedures mentioned above to update the flow discharge, the water depth in each grid is computed with equation (5). Considering the parsimonious demand for input data

and the lower computational burden, the SGC model is a promising hydrodynamic model. In the next section, several tests are configured to evaluate the computational and numerical performance of the new SGC model.

## 3 Model testing and results

A structured sequence of numerical experiments that provide a rigorous test of the numerical and computational performance of the enhanced SGC model is configured. The accuracy and efficiency of the new SGC model is evaluated against the original

SGC model (Neal et al., 2012), a first-order finite volume (fv1) solver (Shaw et al., 2020), and a second-order discontinuous Galerkin (dg2) solver (Kesserwani et al., 2018; Shaw et al., 2020) within the same framework (LISFLOOD-FP). Realistic uncertainties over topographic errors and model system uncertainties can be compensated for by using the same code structure, thus making direct comparisons between the different techniques easier to achieve. All four solvers can be activated by assigning a separate parameter, which is very convenient for performing the calculation and comparisons. The original SGC

model applied the solution scheme proposed by Bates et al. (2010) to discretize the local inertial equations. It has been reported that the scheme tends to break down for urban areas where the Manning values can be less than $0.03\text{m}^{-1/3}\text{s}$. The fv1 solver is a first-order finite volume method solving the integral form of full shallow water equations (SWEs), and discontinuity in the solutions is captured by the HLLC approximate Riemann solver. The dg2 solver applies the discontinuous Galerkin method to solve the full-dynamic system (Kesserwani, Ayog and Bau 2018). A detailed description of these models

can be accessed from this paper (Shaw et al. 2021). In principle, dg2 is the most accurate 2D hydrodynamic solver mathematically among the four, and a high computational resource is required to attain a second-order accuracy. The main features of the four solvers are listed in Table1. These models are set up to cover the potential theoretical limitations of the local inertial equations (de Almeida and Bates, 2013; Cozzolino et al., 2019), especially for areas where complex flow patterns are easy to form, involving rapidly varying, supercritical flows, shock waves, or flows over very smooth surfaces.





**Table 1: Main features of solvers in LISFLOOD-FP model.**

| Model | Equations | Shock capturing | Time step and Stability | Computational efficiency | Accuracy | Disadvantages |
|---|---|---|---|---|---|---|
| **Original SGC** | Local inertial equations | Shocks are not represented by the governing equations. | $t = cfl\dfrac{dx}{\sqrt{g\mathrm{h}_{max}}}$ Unconditionally stable | High efficiency based on OpenMP. | Well-approximated for subcritical and gradually varied flow. | The scheme tends to be unstable for urban areas (n<0.03). |
| **Improved SGC** | | | | | | The scheme may overestimate the water depth near the wavefront for high friction (n>0.06). |
| **fv1** | Full 2D shallow water equations | The shocks are captured by Godunov's method. | t= $cfl\dfrac{dx}{|u|+\sqrt{g\mathrm{h}_{max}}}$ Unconditionally stable with 0<*cfl*<0.33. | A high computational resource is required. | Less accurate | Fv1 solver would in general overestimate the water depth. |
| **dg2** | | | | | The most accurate solver theoretically. | Time-consuming. A small-time step is demanded. |

The fv1 and dg2 solver are full 2D hydrodynamic models without specially designed river routing schemes. River

hydrodynamics can be achieved with a high-resolution grid to include the predefined bathymetric features, where the channel width and depth data are burned into the DEM in advance. Considering the variety of spatial discretization of fv1 and dg2 used, the first priority is to make sure that assimilation of the elevation data into each model leads to an identical representation of the site terrain. For an idealized test where river channel width is uniform, a high-resolution grid can be utilized to characterize the river channel, and the elevation data can be tailored manually before conducting

the simulation to incorporate bathymetric information. Specifically, the strategy is to set the grid dimension identical to uniform river channel width and reduce the DEM elevation to the channel bottom for both the fv1 and dg2 solvers. For the remaining real-world tests, the limited grid resolution to identify the small-scale river channel because of the heavy computational burden makes it impossible to include the river bathymetry in full 2D models. The fv1 and dg2 solver are here used to provide inundation extent without considering the river hydrodynamics in order to analyse whether it is

necessary to incorporate the sub-scale river component in flood modelling. In-situ observation data are provided to benchmark the model performance. Specifically, these tests are:

Test 1: non-breaking wave run-up on a planar beach

Test 2: flow discharge distribution at a river junction

Test 3: subcritical flow over an undulating bed elevation in a rectangular channel

Test 4: simulation of flood propagation through a complex street and building network at a fine spatial resolution

Test 5: simulation of flood propagation in an integrated system with complex river channel network and floodplain topography

Tests 1 and 3 are designed to explore the model stability and accuracy for a steady-state and an unsteady problem respectively, each with an analytical solution or a well-approximated solution provided. Here the model ability for wave propagation in the





river channel is assessed without considering the overland flow inundation process. The modelling performance at a river
junction is assessed in test 2, to validate the accuracy and robustness of the adaptive upwind solution scheme there, and assess
how the scheme performs at distributary junctions. Test 4 is designed to test pure floodplain inundation over a complex urban
area, with the emphasis on the assessment of the availability of the simplified model for the area where the complex flow
pattern exists. The overall performance is assessed in tests 5, which involve the simultaneous calculation of river
hydrodynamics and floodplain inundation process. Tests 1-3 were run on an Intel core i7-10850H 8-core CPU (@ 2.70 GHz)
with 16 GB of main memory and an OpenMP parallelization strategy, and tests 4-5 were run on the University of Bristol HPC
system with 4 nodes of a 4-core processor with 64 GB RAM per node. Previously, these tests have been successful used in
practical applications. By conducting these tests, the model ability to simulate the wave propagation in the river channel, flood
inundation over the floodplain, and water exchange over the river channel and floodplain boundaries is investigated. The model
performance is assessed in terms of the RMSE, volume error, and total computational time.

**Test 1: non-breaking wave run-up on a planar beach**

The test developed by Hunter et al. (2005) aims to simulate the water flow running up a planar beach as the upstream water
depth rises slowly. No analytical solution exists but a well-approximated result can be acquired by the 4th Runge-Kutta method
and taken as the analytical solution. The planar beach with an adverse slope of 1/6000 is configured with a wide range of
friction parameters (0.01, 0.03, and 0.06m$^{-1/3}$s) to check if the improved SGC model can provide stable solutions, especially
for low friction areas where the original SGC code is prone to collapse. The flow speed is u = 1ms$^{-1}$, and mesh resolution is
50m and the total simulation time is 5000s. The test reveals key details about the fundamental capabilities of the solvers in
simulating the propagation of flood waves across initially dry sloping beds.

All four solvers run efficiently for the planar beach test with such a simple topography and boundary conditions, and the results
are all acquired within 1 minute, most of which are fixed compute costs such as I/O and array initialization. The dominant
wave propagation characteristics on the initially dry bed sloping planar can be captured by all four solvers. Considering the
accuracy of the four solvers, the dg2 solver based on the full SWEs has the best approximation to the reference solution, while
an overestimation of the water depth is captured by the fv1 solver in all three scenarios (Figures 2 and 3). Even though the fv1
solver is constructed on a full-dynamic system, it does not perform as well as the improved SGC model based on the local
inertial equations. A possible reason why the fv1 solver has an unsatisfactory performance is that the first order finite-volume
approximation cannot keep all the necessary flow information due to the truncation error. The two SGC solvers have some of
the properties of a second-order scheme as a result of the staggered grid and therefore avoid this issue. While the dg2 solver
based on the full-dynamic SWEs is preferred here due to its high accuracy, an efficient solver based on the simplified version
governing equations that retains the dominant hydraulic features of the flow field would be favoured for many practical
applications. The improved SGC model appears to be a promising solver with less computational resource demand than dg2
considering that the maximum volume error in all three scenarios is -0.5%.





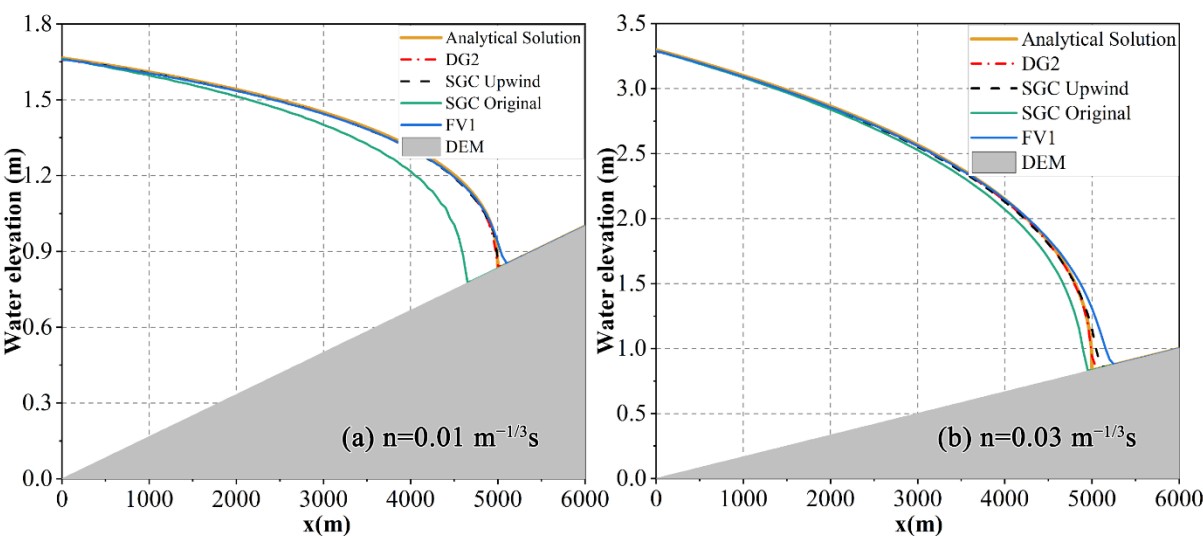

**Figure 2: The water elevation profile calculated by the four solvers (dg2, fv1, original SGC and improved SGC) under n = 0.01 and 0.03m−1/3s at t = 5000s.**

As depicted in figure 2(a), the original SGC scheme deviates from the others as expected at $n=0.01\text{m}^{-1/3}\text{s}$, while the improved

SGC model with the upwind solution scheme is still accurate at this friction. The fluctuation in the water elevation profile at low friction also implies the original SGC scheme has broken down here and become unstable. A tendency for deviating from the analytical solution can be found under $n=0.03\text{m}^{-1/3}\text{s}$ (Figure 2(b)) where the original SGC model has a slow wave speed (17% slower than analytical solution near the wave front), and the predicted water depth follows behind the analytical solutions. The difference grows as the friction parameter progressively reduces, and eventually the original scheme diverged, resulting

in a 279m lags at the wave front after 5000m of wave travel. The divergence is determined by the intrinsic features of the discretization method used in the original SGC method. After implementing the upwind solution-sensitive stencil, the artificial diffusion which coincidences with the direction of flow information propagation is included, and a more robust and accurate stencil is formulated. The improved SGC model outperforms the original SGC model, and the volume error and the RMSE from the analytical solution also evidence the better performance of the improved SGC model (Table 2).

The improved SGC model does not always have a better performance than the original scheme over the whole domain. For the simulation with a high friction parameter ($n=0.06\text{m}^{-1/3}\text{s}$), a slight overestimate of the water depth is found near the wave front for the improved SGC model, shown in Figure 3. Even with the divergence near the wave front, RMSE for the improved SGC model is still only 0.07 m. For many applications this is probably acceptable as this is less than the vertical error in typically available terrain data (Bates et al., 2010). Considering the overall performance of all the solvers, the new SGC solver

indeed improved the stability and accuracy of the original scheme, especially for the low friction case.



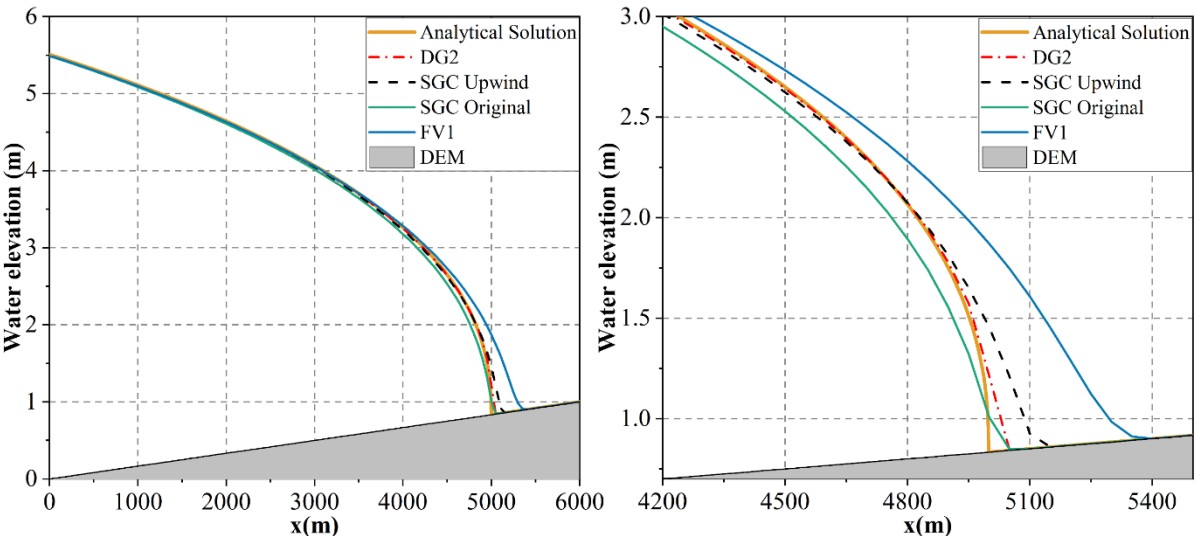

**Figure 3: The water elevation profile calculated by the four solvers (dg2, fv1, original SGC and improved SGC) under n = 0.06m−1/3s at t=5000s and an enlarged view at the waterfront.**

Table 2 shows the volume error and the RMSE of the water elevation from the analytical solution of the four solvers under

different friction scenarios. A sufficiently physical description of the wave run up on an adverse slope is captured by the dg2 solver, and the predicted water depth is a good approximation of the analytical solution, with the maximum RMSE of 0.0374m and maximum volume error of -0.48% in all three scenarios. The total volume of water in the domain is error-free even after a long-time evolution of the full-dynamic system. The improved SGC model shows a comparable performance to the dg2 solver, with the maximum RMSE of 0.0715m. The original SGC and the fv1 solvers are more sensitive to friction parameters.

An obvious underestimation of the water depth is witnessed at n=0.01m$^{-1/3}$s for original SGC with the maximum volume error of -6.2588%, while a significant overestimation of the predicted water depth is observed for the fv1 solver with maximum RMSE of 0.174m. The fluctuations in model performance for the original SGC and fv1 solver indicates that the solution method is of vital significance compared with the different versions of the governing equations. Given the overall performance of the improved SGC model, we can confirm that the upwind form stencil with adaptive artificial diffusion enhances the model

stability and accuracy over the original SGC scheme.





**Table 2: Impact of the friction parameter on water elevation RMSE and volume error**

|  | n=0.01 | | n=0.03 | | n=0.06 | |
|---|---|---|---|---|---|---|
|  | RMSE (m) | Volume error (%) | RMSE (m) | Volume error (%) | RMSE (m) | Volume error (%) |
| **dg2** | 0.0062 | -0.4804 | 0.0134 | -0.1083 | 0.0374 | -0.1990 |
| **improved SGC** | 0.0066 | -0.5063 | 0.0349 | -0.3580 | 0.0715 | -0.3920 |
| **original SGC** | 0.0936 | -6.2588 | 0.0784 | -2.5186 | 0.0657 | -0.3202 |
| **fv1** | 0.0113 | -0.6704 | 0.0666 | 0.9555 | 0.1740 | 1.5814 |

**Test 2: flow discharge distribution at a river junction**

A reasonable approximation of the flow discharge distribution at a river confluence is of vital importance for river hydraulics modelling in many situations. Modelling performance in a single river channel is validated in Test 1, while the reliability and the accuracy of the new solution scheme at a river confluence has not been evaluated in any previous research, and little knowledge is known on how the scheme performs at distributary junctions. Test 2 is therefore configured to assess the robustness of the new river hydraulics solution scheme in this situation. A total of three cases with different arrangements of

a rectangular river channel, ranging from simple to complex, are set up for evaluation of the river hydraulics solution strategy. As shown in Figure 4 (a), a sloping rectangular river channel with a fixed water head of 0.1m (i.e., steady state conditions) is set up, with a slope of 1/1000 and a uniform Manning parameter of $0.06 \text{m}^{-1/3}\text{s}$. The same configuration also applies to case (b) and (c). Case (b) combines two identical rectangular river channels, intersecting at the midpoint of these two channels. River flow with the same water depth boundary condition as in test (a) in these two tributaries runs down at a planar slope and

joins at the river junction cell, and the intersection cell plays the role balancing the discharge allocations for the two downstream tributaries. As the modelling accuracy in the river channel has been validated in test 1, case (a) is taken as a reference for the validation of the solution accuracy at a river confluence and no analytical solution is provided. Case (c) is configured to evaluate the discharge distribution at a river junction, where water is continuously supplied from one main stream. All the river channels are surrounded with a bank that is sufficiently high to avoid water exchange at the river-

floodplain interface.





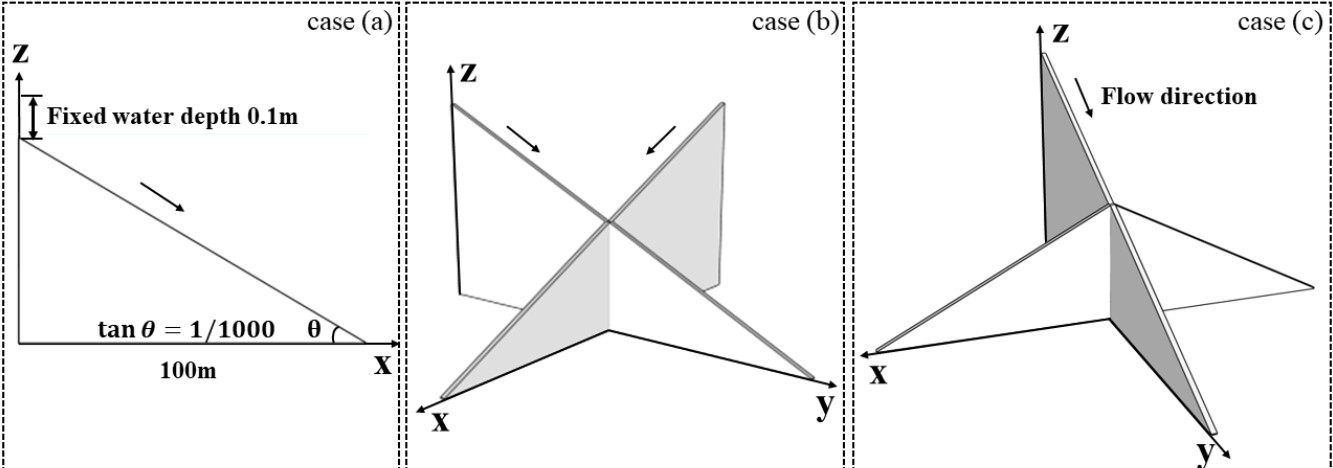

**Figure 4: Validation of the new solution strategy at a river junction with different arrangements of a sloping rectangular river channel. (a) a single river channel. (b) two identical sloping channels intersecting at a junction cell. (c) a main stream provides flow for three same downstream channels.**

For the calculation of the local cell surface flux at one boundary of a river confluence grid, all the other three grid boundaries may be included, depending on the local flow direction, as all flux exchanges with the confluence cell are responsible for the mass balance there (equations (8) and (9)). The water depth profiles for these three cases at t=131s are shown in Figure 5, representing a transient flow status before reaching a steady state solution, which is a common water depth profile in this situation. A similar slight overestimation of the water depth near the wave front as in test 1was found for the upwind solution scheme (blue line) in case (a), with a Manning parameter of $0.06\text{m}^{-1/3}\text{s}$. An identical water depth profile is calculated for the two downstream tributaries in case (b), marked as purple line and circles in Figure 5, which guarantees that an identical treatment is applied to the horizontal and vertical flow movement in the new upwind solution schemes. The near identical water depths in case (a) and (b) confirms that an acceptable performance has been achieved with the new scheme. The slight overestimation of the water depth in case (b) is influenced by the intersection of the two streams. When the two streams join at the river junction cell, we have twice the discharge entering the cell without quite twice the cell area. The water volume at the intersection cell is increasing and a higher water depth gradient can be expected for the two downstream tributaries. Therefore, A greater flow velocity exists due to the large water elevation gradient, resulting in an expected overestimation of the water depth profile compared with the single river channel case. Case (c) demonstrates the discharge distribution for three downstream tributaries. A symmetrical water depth profile in the y direction (red line), which is the same as the water depth in the x direction (blue circle), is acquired, implying that the discharge is evenly allocated between the three tributaries. Inclusion of all upwind exchange which may have potential impact on the mass balance at a river junction cell guarantees the identical treatment of these downstream tributaries. If only one upwind discharge in the same direction is considered, water depth in the x direction can be always higher than the y direction as no upwind discharge exists for the flow in the y direction at the intersection of these three tributaries.





Assuming that no energy loss is generated when flow direction changes, the new upwind solutions schemes keep a balance between the mass storage and the momentum exchange at a river confluence, making sure that the flow movement can be accomplished in a simplified way.

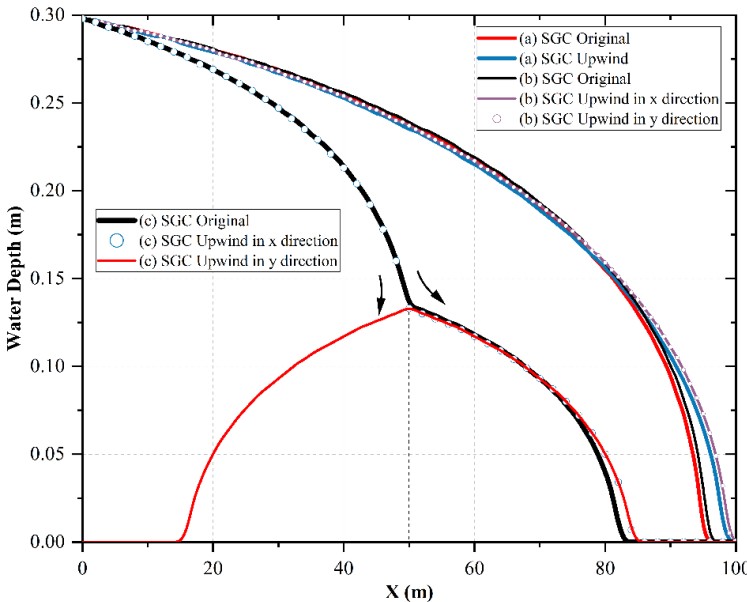

**Figure 5: Water depth profile predicted by the two SGC solvers at t=131s. Group (a) shows the water depth predicted by a single sloping river channel, and (b) is the 2 channels intersecting at a confluence, and (c) shows the flow distribution from a main stream to three downstream channels.**

**Test 3: subcritical flow over an undulating bed elevation in a rectangular channel**

Test 3 is derived from de Almeida and Bates (2013) and explores a steady flow regime in a 5km long and 10m wide rectangular channel with a Manning coefficient of $0.03\text{m}^{-1/3}\text{s}$ (a 1-km-long river is also configured). The upstream discharge is set to $2\text{m}^3/\text{s}$ and the downstream water elevation is determined according to equation (14). To derive the analytical solutions of the steady flow regime, a subtle approach is used for the inverse problem. Given the water elevation profile and boundary conditions, the bed elevation that we seek is acquired by integrating the channel slope analytically derived from the steady flow form of the Saint-Venant equations. This test provides a rigorous assessment of the model's ability to accurately simulate a range of steady-state flow conditions.

$$S_0 = \left[1 - \frac{4}{gh(x)^3}\right]h'(x) + \frac{0.0036}{h(x)^{10/3}}, \tag{13}$$

$$h(x) = \frac{9}{8} + \frac{1}{4}\sin\left(\frac{\pi x}{500}\right), \tag{14}$$

$$h'(x) = \frac{\pi}{2000}\cos\left(\frac{\pi x}{500}\right), \tag{15}$$





Where $h(x)$ is the water depth profile and $h'(x)$ its spatial derivative, $S_0$ is the channel slope. Following equations (13)-(15),

the analytical solution of the water depth and channel bed elevation for the 5-km-long channel is formulated. Due to the advection term being ignored in the local inertial equations, it is considered that the side effect of the simplified local inertial equations is to attenuate any oscillations of the water depth (De Almeida and Bates 2013). Therefore, the 1-km-long channel is configured to introduce a more oscillating channel bed elevation, investigating the applicability of the local inertial equations for a high-oscillation environment. To set up the analytical solution for the 1 km channel, the wavelength in equation (14) is

reduced by a factor of 5 while the wave number remains unchanged, and thus the oscillation frequency increases. The water depth gradient is computed by differentiating equation (14). After substituting equations (14) and (15) into equation (13), the slope along the river channel is determined, and the channel profile is calculated by further integrating the slope using high-resolution quadrature methods. More simply, the 1-km-long channel is achieved by compressing the x-axis of the 5-km-long channel, whilst the wave amplitude remains unchanged. A comparison of the analytical solution with the water depth and

channel bed elevation calculated by the different solvers is shown in Figure 6(a). The predicted water depth from all the four solvers agrees well with the analytical solution at the final steady-state, with maximum RMSE of 0.0213m. The water oscillates at a high frequency in the 1-km-long channel, and a highly oscillatory water depth profile is therefore created. However, an opposite trend is witnessed in Figure 6(b) for solvers based on the full SWEs (fv1 and dg2) and the local inertial equations (SGC solvers) as we reduce the channel length.

As is depicted in Figure 6(b) and (c), the water depth predicted by the local inertial equations is lower than the full-dynamic SWEs for subcritical flow, indicating that the SGC model indeed attenuates some oscillations in the water depth. As we decrease the channel length and naturally increase the frequency of the channel bed oscillation, an increased departure from the solution is obtained for models based on the different governing equations. Owing to the impact of the nonlinearity in the friction term, a larger energy loss is included for the highly oscillatory depth profile. As a consequence, a further

underestimation of the water depth is found in the SGC model for the 1-km-long channel. Originating from the intrinsic characteristics of the governing equations, the discretization method cannot change the final behaviour of the steady-state problem. At last, the water depth calculated by the two SGC solvers shares the same water depth profile, even though they are different before the formulation of the steady-state. While for the fv1 and dg2 solver, the highly oscillatory depth profile is also accompanied by greater energy losses without the attenuation of the flow depth gradient. The water depth is retained by

the full-dynamic system, and a high-fluctuation water depth profile is well-captured by the solvers based on the full-dynamic system. However, the disparity is relatively small (0.07m) compared to even the best wide area terrain data (i.e., airborne laser altimetry or LiDAR), indicating that the SGC model based on the local inertial equations could still be used for a wide range of steady-state problems.

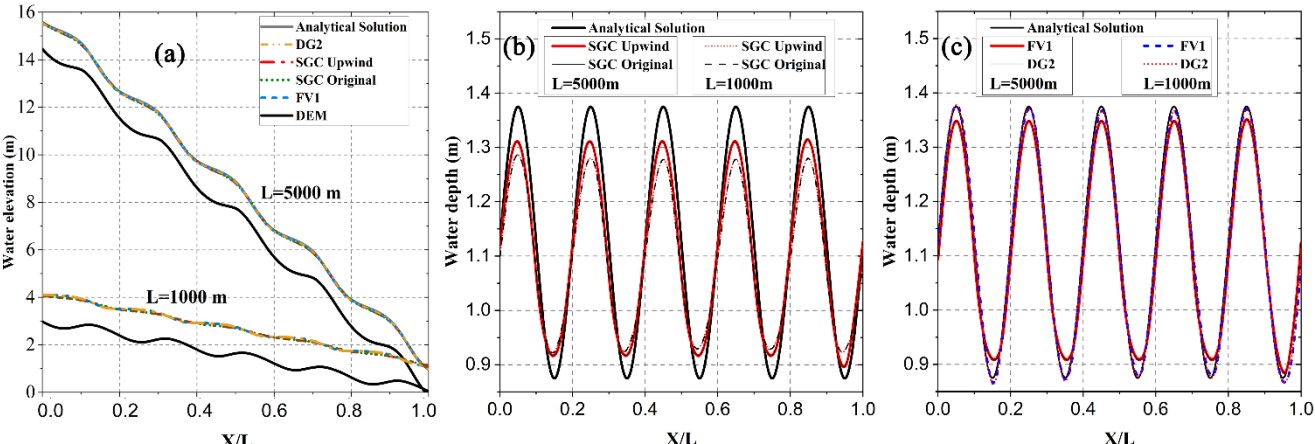

**Figure 6: Water depth profile predicted by four solvers under L=1000 m and L=5000 m. (a) water elevation and the river channel bed elevation profiles. (b) water depth predicted by two SGC solvers under different oscillating frequency. (c) water depth predicted by the fv1/dg2 solvers.**

It takes about 2 minutes for the SGC model to achieve the steady-state water depth profile for L=5000 m, which is 5× faster than the dg2 solver and 2× faster than the fv1 solver. Considering the lower computational resources required while still preserving spatially second-order accurate results that are close to the dg2 solver given typical real-world errors, the SGC solver would be a promising alternative especially for large-scale flood modelling. From the perspective of model accuracy, all numerical results show an excellent agreement to the analytical solution when they reach steady state. However, it has been noticed that the original scheme struggles to keep a stable state, and divergence of the original SGC scheme is captured before reaching the final steady state, even though a high-resolution 1-m grid is adopted. While for the improved SGC model, no obvious divergence is captured during the formulation process of the steady-state. By implementing the improved numerical scheme in the SGC model, the potential instability problem can be eliminated, enabling the achievement of a reliable result more easily.

The adaptive weighting factor $\theta$ in equation (6) has a minimum value of 0.3, provided that the $cfl$ in equation (12) is 0.7. Statistics of the of the distribution of $\theta$ acquired during the calculation process indicates that ~92% ranges from 0.70 to 0.83, only at the very early beginning of the simulation the $\theta$ can be less than 0.7. After reaching a steady state, nearly all $\theta$ remains within 0.70-0.83. Tests 1 and 2 also witnessed that the $\theta$ can be smaller than 0.7 only in very rare conditions. Therefore, an extra constraint is adopted, which allows that the $\theta$ varies from 0.7 to 1.0. A similar strategy has also been implemented in Sridharan et al. (2021). By limiting the minimum value of $\theta$ to 0.7, limited impact of upwind flow information can be applied, which makes sure that the local flow status can always dominate the local discharge update. Without such a constraint, the dominant upwind flow discharge can always accelerate the flow speed, and too much dependent on the upwind flow information while ignoring the local flow status can easily cause mass balance error.



**Test 4: fine resolution flood propagation over the complex urban environment**

Tests 1-3 are idealized cases that assess the accuracy and stability of the improved SGC model against analytical solutions.
However, the model ability and efficiency for modelling river-floodplain systems has not yet been evaluated. The following

tests are configured to test the simulation of real-world flood propagation process to thoroughly check the model performance.
Test 3 is motivated by Hunter et al. (2008) and is applied to investigate the model ability to simulate flood propagation over
complex topography. The domain covers an area of 1000 m by 400 m in the city of Glasgow, Scotland, UK, with dense urban
development along both sides of the two main streets at the site and a topologically dense network of minor roads, as shown
in Figure 7. A flash flood event with rapid hydrograph rise and fall that occurred on 30 July 2002 is simulated. A 1 m resolution

LiDAR DEM is filtered to remove the buildings and vegetation to give a "bare earth" DEM that includes some steep stretches
of road and isolated depressions where water may accumulate. The DEM has horizontal and vertical accuracy less than 50 cm
and    15    cm    RMSE    respectively    and    is    further    fused    with    the    Ordnance    Survey    OpenData
(https://osdatahub.os.uk/downloads/open) to identify the location of the buildings. The cells where the buildings are located
are raised in elevation by 6 m to represent an arbitrary building height, as shown in Figure 5. A point source inflow boundary

condition with the same hydrograph as Hunter et al. (2008) is established at location P0, and other boundary conditions are set
to closed since there is no mass flux interaction at these boundaries. The flash flood lasts about 1 hour, and a simulation period
of up to 5 hours is conducted to fully capture the developed flow field. A Manning parameter of $0.02\mathrm{m}^{-1/3}$s is selected for the
streets as these have a smooth surface, while other areas adopt n=0.05 $\mathrm{m}^{-1/3}$s friction coefficients to represent the uneven
terrain topography.

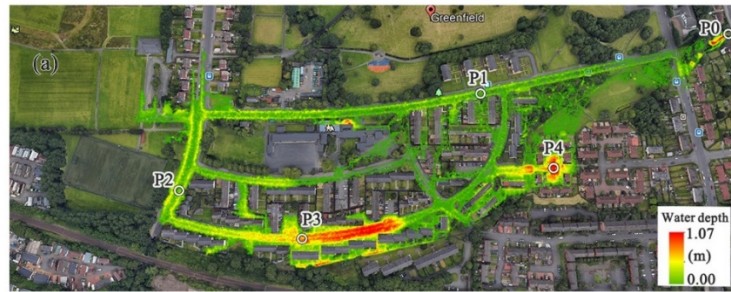

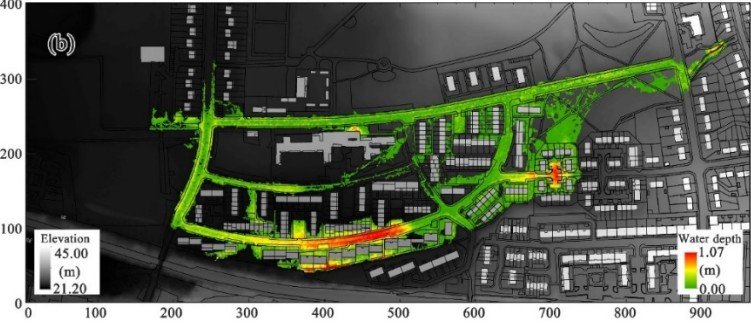




**Figure 7: Flood inundation extent at t = 4200s calculated by (a) dg2 solver and (b) the improved SGC solver. Background map in (a) are © Google Maps screenshot(s).**

Figure 7 illustrates the predicted water depth distribution of the dg2 and improved SGC solvers. A similar flood inundation extent is obtained by these two solvers, while the dg2 predicted a later recession of the flood propagation. The interaction with the building configuration and convergence in low-lying regions characterizes the complex flow field. Flow is thus a complicated combination of high-velocity shallow flow and ponding in low-lying regions. The complex flow patterns (e.g., numerous transitions to supercritical flow, numerical shocks, and flows over very smooth surfaces) in the urban environment pose significant challenges for an accurate representation of the flood inundation process. As we can expect, the SGC model cannot capture all the hydrodynamic features in such an environment while only retaining the dominant factors impacting the flood propagation (pressure, friction and local acceleration). However, the improved SGC model is found to provide a reasonable approximation to the full-dynamic shallow water equations (SWEs) solvers considering the water depth distribution. At peak inundation, the mass error is comparable to a height error of only 3.5mm when dispersed across the whole inundated region. When it comes to the computational efficiency, the dg2 solver and fv1 solver have an increase in the simulation time compared with the SGC model. The SGC solver acquires the final results within two minutes, which is 4× and 6× faster than the fv1 and dg2 solver respectively.



**Figure 8: Maximum water elevation difference out of the fv1 and SGC solvers compared with dg2. Background vector data are © Ordnance Survey OpenData.**

Owing to the lack of in situ observation data, results calculated by the dg2 solver are taken as the benchmark. The at least second-order accurate dg2 solver can acquire more details impacting the hydrodynamic processes, leading to a more complete water depth distribution. Figure 8 shows the maximum absolute error distribution of the fv1 and SGC solvers compared with the dg2 result. Considering that all of the absolute errors are within the range 0.1m, the specific error value is omitted, and only the three categories characterizing the maximum deviation from the dg2 results at each DEM cell are depicted. 57% of the area is occupied by the red colour, which indicates that the water depth difference between the original SGC solver and the



dg2 is the largest. In the southern part of the domain where the low-lying regions suffer serve from flood damage, the fv1 solver gives the largest overestimatsion of water depth (relative to dg2), with the areas where fv1 are maximal occupying 39% of the whole inundation area. The improved SGC solver yields well-approximated results compared with the dg2 solver, only 4% of the inundation extent shows an obvious deviation from the dg2. One possible reason why the new SGC solver can improve the model accuracy compared to the original SGC scheme is that the latter is going to break down in such an urban

environment, especially for areas with a combination of small Manning value and shallow water depth. The implementation of the upwind scheme together with the adaptive weighting factor alleviates the tendency of divergence. The improved SGC solver predicted a slightly larger inundation extent near P0, illustrating a rapid ponding area at the start of the simulation, followed by a gradual release of water as the simulation progresses. The upwind water depth is always higher here, and the upwind scheme would predict a slightly larger surface flux compared with the scheme adopted in the

original SGC solver. Thus, the flow can march further at the wet-dry boundary, and a broader inundation extent exists. In other areas of the domain the different flow tributaries and the interaction with the buildings induces a varying flow field. As a result, the predicted surface flux from the new SGC solvers approaches the true flux as a result of the adaptive weighting factor. The proposed SGC model (only floodplain component as no channel is included in this test) can provide oscillation-free solutions even on smooth surfaces in urban areas, and exhibits a good agreement with dg2 in most

locations without the need for trial-and-error modification of the value of the diffusion coefficient.



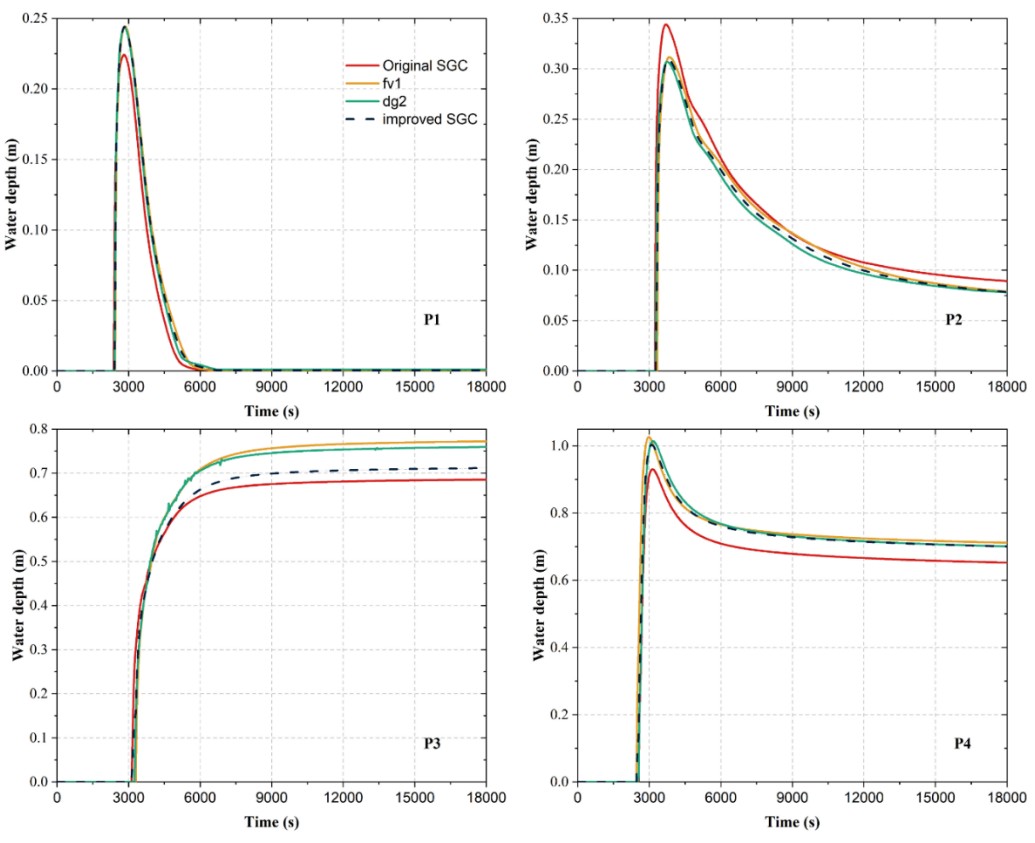

**Figure 9: Time-series of water depth variation in four locations: P1-P4 (locations of P1-P4 are shown in Figure 7).**

Figure 9 depicts the time-series water depth variation at the four points shown in Figure 7. P1 is monitoring the water depth variation on the street close to the water inlet. The relatively smooth street transfers the flood wave downstream quickly without

holding back the water volume. The shallow water begins to divide into several branches as the flow proceeds further into the domain. The original SGC solver predicts an early arrival of the flood peak, together with a quick recession as the simulation proceeds. The maximum deviation of the flood peak is 0.169m, and the water depth predicted by the other three solvers is in good agreement. Data from P2 shows that the original SGC solver attenuates the water amplitude as the flow is shallow and high-velocity status on the street. A slight overestimation of the flood peak compared to the others is captured by this solver,

and the discrepancies are about 4cm which is of the same order as the vertical error in the LiDAR DEM (RMSE of ~5cm). P4 is an area where the flow interacts with buildings, and the combination of shallow water depth and the flow around building blocks leads to a complex flow status. As a result, the RMSE between the original SGC and dg2 is ~5cm, while a 1.5cm RMSE is given for the improved SGC solver. Complex flow patterns also exist near the P3. The high-speed shallow water from both sides of the street merges, forming a deep low-velocity flow field. The constraints imposed by the buildings on both sides of

the street influences the flow propagation direction, which further aggravates flow oscillations. Therefore, a large volume of





water converges here, leading to higher water depth and obvious water depth differences. However, the maximum water depth difference is limited to within 0.1m. The original SGC solver still has the maximum deviation from dg2, with a maximum RMSE of ~6cm, while the new SGC solver has an RMSE of ~3.5cm. Overall, the improved SGC model can generate oscillation-free solutions even on smooth surfaces in urban areas and shows a good agreement at all stations to a full shallow

water second-order model.

**Test 5: Simulation of flood propagation in an integrated system with complex river channel network and floodplain topography**

Test 5 reproduces a flood inundation process in the Carlisle 2005 urban flood event caused by heavy rainfall. Over a 36-h period preceding the flooding, up to 175mm of rain fell over the Eden catchment in which the city of Carlisle sits. Overflow

from the River Eden and backwater flow impacts along the Rivers Caldew and Petteril aggravated a severe situation and led to considerable flood damage. Overall performance of the improved SGC model is assessed in test 5, and the modelling capacity to represent wave propagation in a river channel with floodplain inundation dynamics and water exchange at the river-floodplain interface are evaluated with the real-world test.

The underlying topography is constructed with a combination of LiDAR fused with digital map data. LiDAR data with the

estimated RMSE of 0.197m are provided by the Environment Agency of England and Wales (EA), as well the cross-section data with an interval of 200m on the River Eden and 50m along the Rivers Petteril and Caldew. Vegetation is removed from the source data while the building information is retained, and thus a 10m resolution DEM with a vertical RMSE of 0.38m and a mean error of 0.07m is acquired. Cross-sections data are further interpolated to approximate the river bathymetry, with the river channel being 'burnt' into the DEM. The resultant topography data with a resolution of 5m is collected from Horritt et

al. (2010). Due to the relatively high runtime cost, the finest 5m DEM is resampled to 10m and a total of ~146 thousand grids are generated. Both fv1 and dg2 are pure 2D models and therefore represent the channel and floodplain as a continuous unit, and these models are therefore applied directly to the 10m DEM with 'burnt in' rivers. In contrast, the SGC model applied an separately configured river channel bathymetry for the river hydraulics calculation, and the 10m DEM are applied for the flood modelling, to ensure that all values inside the SGC output stencil are equivalent to the fv1/dg2 model result with no averaging

or interpolation. The optimised Manning parameter is $0.04\text{m}^{-1/3}\text{s}$ for the channel and $0.06\text{m}^{-1/3}\text{s}$ for the floodplain. The same time series discharge as in Neal et al. (2009) is taken as the inflow for the three rivers, and a free outlet boundary condition is imposed on the River Eden close to the Sheepmount gauging station. All flow is expected to enter/leave the model domain via these river channels. Two post-event surveys of the wrack and water marks were undertaken by the University of Bristol and the EA, and 183-point measurements of maximum water surface elevation were acquired. Given the major aim of the test is

to evaluate the model performance, the model still employs the 2005 terrain and flood defence information even though new defences were constructed following the 2005 flood event.


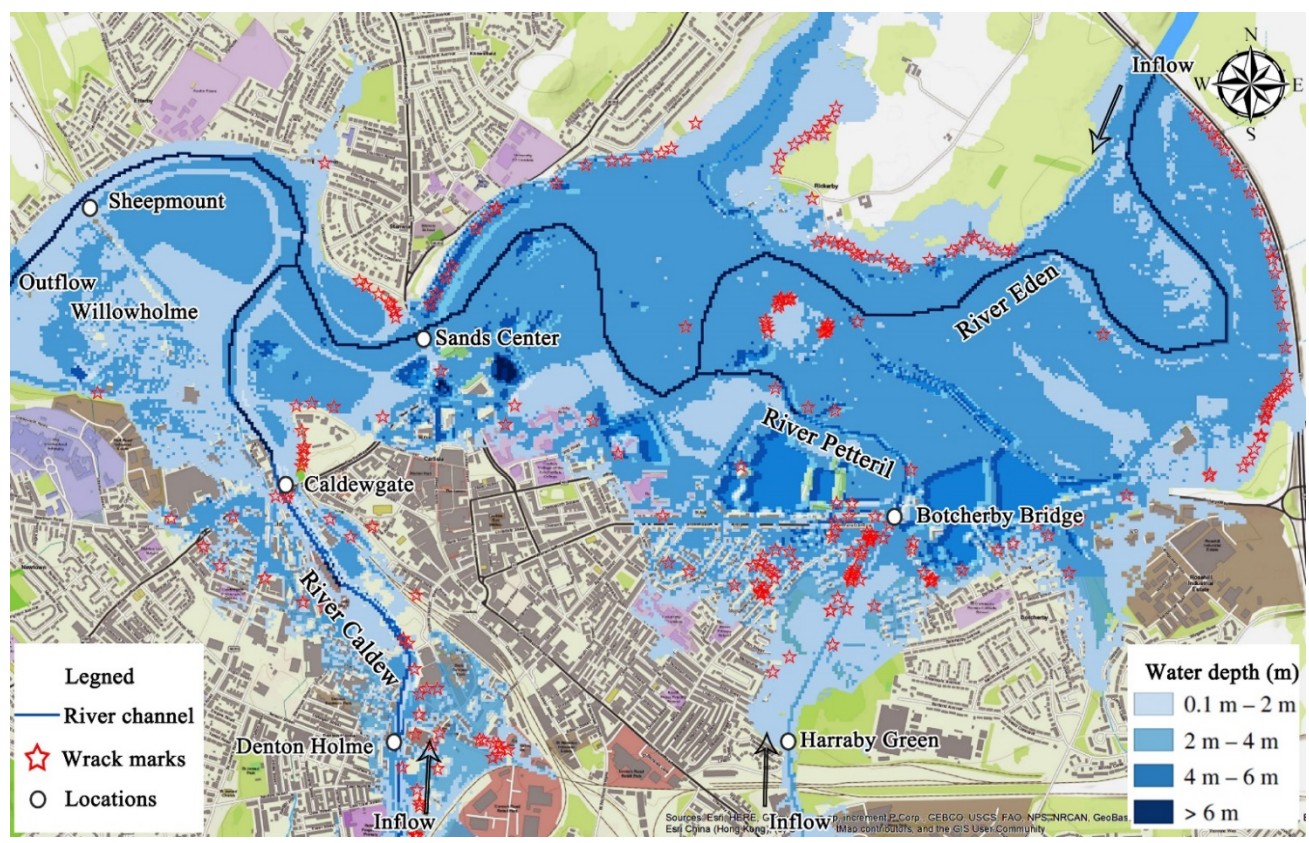

**Figure 10: Water depth distribution for Carlisle calculated by the improved SGC solver at a resolution of 10m. Background maps are © Google Maps screenshot(s).**

The maximum flood inundation extent with the water depth distribution calculated by the improved SGC solver is depicted in Figure 10. Many surrounding districts of Carlisle such as Willowholme, Caldewgate, Denton Holme, Botcherby and Harraby Green are flooded, and this coincides with the observed flood inundation extent. The dark blue line with a water depth deeper than 6m characterizes the locations of the three major rivers in the domain. Further analysis of the inundation propagation process confirmed that the river channel conveys a significant proportion of the flood volume, and at the end of the simulation

67% of the flood volume is routing downstream through the river channel at the outlet boundary. This test highlights the vital importance of in-channel hydraulics modelling even during out of bank floods. The modelling of flow conveyance capacity and estimation of mass and momentum exchange at the river-floodplain interface impacts the inundation process significantly. Even with a 10m DEM and full 2D model, the river channel wave propagation cannot be correctly represented in this case without an independent river hydraulics model, and a slight different inundation process is obtained with fv1/dg2 that has

larger errors compared to the observed water level data. Floods in the full 2D hydrodynamics model may spread over the floodplain earlier than the SGC solvers, and the final flood inundation extent is decreasing as more discharge is routing downstream by the river channel. While in fv1 model, an over-prediction of inundation extent, as well as a higher water depth distribution, occurs without a dedicated river channel model. Further analysis shows that the topography data quality is




responsible for the performance of the full 2D models as river channel features is smooth out during the downsampling process.

Within dg2 solver, piecewise-planar representations of topography and flow variables are required to capture smooth, linear variations within each DEM grid while simultaneously allowing flow discontinuities. An average over four neighbouring DEM cells and two slope-coefficients in x and y directions is acquired for locally planar representation of topography. The interpolation procedure of the fine DEM data with the relatively small river channel bathymetry can greatly affect the quality of the river channel bathymetry, and as a result river hydraulics calculation in the full 2D model is impacted with the crudeness

of topography data. Therefore even with the finest 5m topography data the river hydraulics cannot be well-represented in dg2 while fv1 can capture a similar wave propagation as the SGC solver at the cost of ~8× computation time of SGC solvers. This test highlight the significance of river channel representation in flood modelling, especially when relative small scale river channel features dominating the flood inundation process.

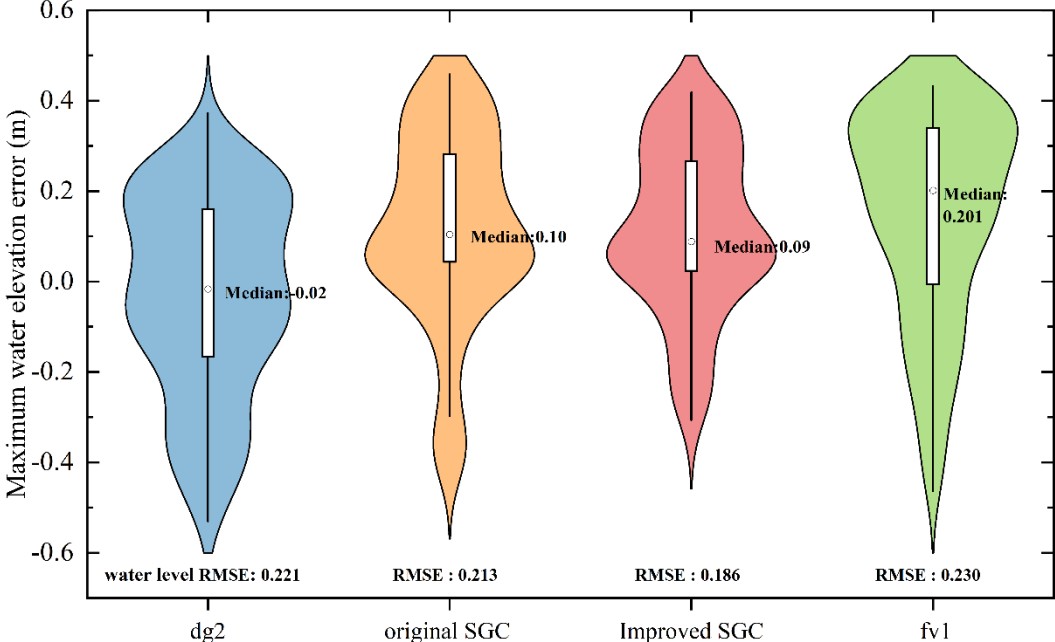

**Figure 11: Water surface elevation errors of the four solvers compared with the field-survey point data.**

Figure 11 shows the predicted maximum water surface elevation errors compared with the field survey data. Some outlying (and likely erroneous) field survey data are excluded since almost all the four solvers show a huge deviation. The errors from the four models are within ±0.5m, with the maximum median value from fv1 of 0.230m. Water surface elevation predicted by the fv1/dg2 solver shows an obvious deviation from the observation data, and an overestimation of the water surface

elevation is widespread over the domain. The RMSE compared with the observed data for fv1 and dg2 is 0.230m and 0.201m respectively. Even though a coarse grid resolution is applied, the improved SGC model predicts a much more reasonable water depth distribution, with a maximum deviation of ±0.5m. Further analysis shows that most of the abnormal points are located





near districts where shallow water interacts with buildings, resulting in a misleading forecast of the water depth distribution. The overall deviation measured by RMSE for the improved SGC is 0.186m, which is the minimum error of the four solvers.

The original SGC solver with the RMSE of 0.213m outperforms the 10m resolution full 2D models. All of the evidence highlights the vital importance of river hydraulics modelling for simulating flood flows.

The final water depth distributions from the two SGC solvers are quite similar, and the maximum difference lies in that the original SGC solver underestimates the water depth with a deviation from in-situ data of -0.4m. Most of these abnormal points are located at the edge of the inundated area. A possible explanation is that the adaptive artificial diffusion solution scheme

can adjust the discharge automatically and responds more quickly to the upwind flow discharge variation, while the original solution scheme is impacted by the small friction and interaction with building blocks and suffers from a too quick flood recession as a result.

Keeping a balance between the modelling efficiency and the grid resolution is a major task for flood modelling, especially when small-scale river channel bathymetry controls the flooding process. Representing dominating river channel bathymetry

features can inevitably increase the runtime of the full 2D hydrodynamic models, while ignoring these features cannot capture the dominant flood inundation process. Strict time step required for full 2D hydrodynamic models stability further limit the modelling efficiency. As a result, the dg2 solver takes more than ~15× computational times than the SGC solver, while the fv1 is ~5.2× slower than the SGC solver. The SGC model with the improved modelling stability in urban environment provides a powerful alternative there, which preserves the small-scale river channel features while keeping a high efficiency. With the

support of the improved SGC model, the model accuracy can be increased to a useful extent. In particular, abnormal water depth distributions can be removed with the adaptive artificial diffusion solution scheme. The improved SGC model therefore provides a better alternative for the river-floodplain inundation simulation.

## 4 Discussion and Conclusions

The SGC model which allows utilization of approximated sub-scale bathymetry while performing computations on relatively

coarse grids has been extensively applied to track the wetting and drying dynamics in the river-floodplain system. Based on the simplified efficient inertial formulation of the shallow water equations, the SGC model provides a feasible solution for large-scale flood modelling. However, the solution scheme of local inertial equations in the original SGC model suffers from numerical instability in the case of low friction scenarios. Many measures have been proposed to improve the accuracy and robustness of the solutions. Unfortunately, the SGC model to date has not included these latest developments in numerical

solutions of the local inertial equations. In this paper, for the first time we implement a previously developed artificial diffusion and explicit adaptive weighting factor in the SGC model. Compared with the original solution stencil, the new solution scheme explicitly includes the artificial diffusion in the form of an upwind scheme to improve the estimation of the numerical flux, and automatic recognition of the diffusion needed to stabilize the solution stencil is achieved with an adaptive procedure based on the local flow status. A further constraint is adopted in this paper to limit the amount of artificial diffusion, which demands





that the adaptive weighting factor varies from 0.7 to 1. Momentum exchange is always dominated by the previous local surface flux, while limited artificial diffusion in the form of upwind surface flux is included, to avoid mass balance errors by this mean. Evaluation of the new SGC model through a structured tests, from simple river hydraulics calculation to real-world flood inundation simulation, confirmed that accurate mass and momentum balance in shallow flows over complex geometries is assured in the presence of wetting and drying. With the inclusion of all upwind surface flux which may impact the mass balance

at a river confluence grid, allocations of discharge between confluence grid and downstream tributaries is achieved under the control of the water depth gradient while ignoring the momentum loss. Momentum loss is also neglected at the river-floodplain interface and this is a reasonable critique of the scheme but implemented for simplicity. All the results, especially the real world tests, indicates that the resulting algorithm is numerically stable, relatively simple and extremely efficient. Together with the separately configured river bathymetry, the new SGC model provides a convenient solution for fine-scale river

hydraulics modelling based on a relative coarse grid, while modelling stability and accuracy is further improved.

Additionally, the examples given demonstrate that the present formulation can generate accurate results, even with a coarse and structured finite difference mesh, and costly and unnecessary grid refinements are avoided with the sub-scale representation of river channel bathymetry. Without compromising the computational efficiency, the new solution stencil improved the model performance in terms of water depth distribution and floodplain inundation extent, especially in the case

of low friction scenarios where abundant smooth urban areas exist. The improved SGC model highlights the vital importance of the representing river flow conveyance capacity modelling and the mass and momentum exchange over river-floodplain boundaries. Without the river channel bathymetry separately included, the full dynamic 2D SWEs solver based on fine resolution DEM data consumes high computational resources while demanding a long time for calculation, and the water depth distribution and inundation extent may not be well presented with a crudeness treatment of the bathymetry in dg2. Furthermore,

the SGC model shows its direct advantage over the full dynamic 2D model in real-world flood modelling. The resource-consuming 2D SWEs solvers demand a relatively small computational timestep which makes them unattractive for flood modelling covering an area up to several hundreds of thousands of square kilometres, while the SGC model with a loose CFL condition can acquire the inundation extent efficiently. Quite different from the fv1 and dg2 solvers, the improved SGC model alleviates the heavy computational burden by including the subgrid scale river channel. The relative average computational

expenses of dg2 solver is ~10× than the SGC solver in real-world test, and fv1 takes 4× more computational time. The river discretization is decoupled from the overlying floodplain grid, and subgrid bathymetry is explicitly included in the solution scheme, permitting a significant gain in efficiency and accurate simulation of the wetting and drying dynamics.  The river hydraulics can be acquired simultaneously with the flood propagation on the floodplain without resorting to costly and unnecessary grid refinements. The results obtained with coarser grid cell sizes and sub-grid sampling are comparable to those

obtained with considerably higher grid cell resolution but at a fraction of the computing effort and data storage.

The adaptive weighting factor in the upwind scheme balances the contribution from the local flux and upwind discharge, with a large value importing less upwind diffusion. The feasible range of the weighting factor is determined empirically, as the upwind solution scheme with a fixed weighting factor is sufficiently stable with a minimum of 0.7. Results from tests 1-3

indicates that a smaller weighting factor can only be acquired at the very early beginning of the simulation, and the factor ranges from 0.70 to 0.83 when a steady stats is achieved in test 3. Therefore, confinement is applied to the adaptive weighting factor which limits its minimum value to 0.70, or a negative volume may be achieved at a river confluence area where three upwind flow discharges are combined for estimating the momentum flux across the cell boundaries. Repeat utilization of the upwind flow discharge for the 4 cell boundaries may result in an overestimation of the output discharge and affect the model mass balance. Relying heavily on upwind discharge while ignoring local flow slope may accelerate the overall wave

propagation speed without any theory evidence. Though an assessment of every cell can be executed to redistribute the discharge once a negative volume is generated, substantial extra computational resources are required. By limiting the minimum value of the adaptive weighting factor, local surface flux are given first priority while updating the momentum exchange for next time step, and limited artificial diffusion impact can propagating upwind flow information while avoiding mass errors. A thorough analytical analysis of the optimized range of the adaptive weighting factor may be conducted in future

work.

In summary, the new SGC model exhibits its advantage over full 2D SWEs solvers in modelling the river hydraulics and floodplain inundation simulation where small-scale river hydraulics has a strong control on the flood generation. With the OpenMP acceleration technology on multi-CPU cores, the SGC model based on the efficient inertial formulation of shallow water equations provides a good approximation to real-world inundation process and shows its great potential in large-scale

modelling. With the adaptive upwind diffusion incorporated, potential instability in the case of low friction scenarios is tackled, and flow conveyance capacity can be modelled with the inclusion of approximate sub-scale bathymetry, providing a compelling alternative for river-floodplain modelling.

**Code and data availability**

Code of the improved subgrid channel model is available on Zenodo (https://doi.org/10.5281/zenodo.7064320), as well as the

test configuration files that can be used for running the original SGC, improved SGC, fv1 and dg2 solvers. Water surface elevation and boundary conditions for Carlisle test are available in Neal, Bates, et al. (2009) at https://doi.org/10.1016/j.jhydrol.2009.01.026.

**Author contributions**

YT coded the numerical solvers under the supervision of PB and JN, conducted simulations and drafted the initial manuscript.

PB and JN gave many suggestions on improving the draft. All authors contributed to conceptualization, manuscript review and editing.



## Competing interests

I declare that I or my co-authors have competing interests as follows: At least one of the (co-)authors is a member of the editorial board of Geoscientific Model Development.

## Acknowledgments

Youtong Rong was supported by the China-Scholarship-Council (CSC) – University of Bristol Joint Ph.D. Scholarships Program. Paul Bates is supported by a Royal Society Wolfson Research Merit award and UKRI Natural Environment Research Council grant NE/V017756/1. Jeffrey Neal is supported by UKRI Natural Environment Research Council grant NE/S015795/1.

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
