# Peer review of "An improved subgrid channel model with upwind form artificial diffusion for river hydrodynamics and floodplain inundation simulation"

_Geoscientific Model Development, 2022_

## Author Response (AR1)

**Response to reviewer 1:**

Thank you very much for your positive comments on the manuscript. Our paper has been revised and checked carefully. Here we reply to these comments from the reviewer point-by-point

*1, Please explain DEM*

**Response:** DEM in this paper refers to Digital Elevation Model, which is a grid-based representation of the bare ground topographic surface in an area. In our improved subgrid channel model, the DEM (bottom boundary conditions) together with the discharge from previous time step controls the discharge exchange in two neighbouring grids, which helps to predict the direction and velocity of water flow during a flood event. Added regarding the DEM that we will define this acronym when first introduced for clarity.

*2, In line 155, "the weighting factor $\theta$ defined in equation (6)". Here it should be equation (4). Again, in line 161, replace equation (6) with equation (4).*

**Response:** Corrected as suggested.

3, In line 180, please replace "gird" with grid.

**Response:** Corrected as suggested.

**Response to reviewer 2:**

We are very grateful to the reviewer for the insightful comments on the manuscript. Their constructive suggestions help a lot on improving the quality of this manuscript. We response the comments from the reviewer point-by-point.

*1, Please clarify how the flow direction is defined at first. Only after defining the flow directions you can say how the negative discharge is calculated, depending on the positive definition of the flow direction.*

**Response:** Corrected as suggested. The flow direction is defined as positive if the flow direction is from west to east or from north to south, with the origin of the domain located at the upper left (northwest) corner. Regardless of the discharge magnitude, a negative discharge corresponds to a flow direction opposite to the positive direction definition. We now introduced the flow direction definition in the method section earlier, and hopefully the meanings of the positive or negative discharge can be much more clearly understood.

*2, The water depth is denoted with h In Equation (4) while R in equation (7). Please clarify what is the difference, and the reasons why.*

**Response:** More explanation has been included in the manuscript to detail the different between these two solution schemes. Equation (4) is the finite difference solution scheme for discharge calculation on the floodplain. $h_{flow}$ is the depth between cells through which water can flow, defined from the water depths and cell elevations z as

$$h_{flow} = \max\left(h_i + z_i, h_{i+1} + z_{i+1}\right) - \max\left(z_i, z_{i+1}\right)$$

Where h is the water depth at the cell centre, and the z is the cell elevations.

Equation (7) is the stencil for river hydraulics calculation. For shallow water flow where the flow depth is far smaller than the channel width, the hydraulic radius can be approximated by the flow depth. However, since the subgrid model could be required to simulate relatively small, narrow, and deep channels, the model formulation was derived for the more general case where the hydraulic radius of the channel is defined as the flow area divided by the wetted perimeter (Neal et al., 2012). The solution scheme of the river flow (equation (7)) therefore uses the hydraulic radius R instead of $h_{flow}$.

*3, How are the Manning parameters selected for the real-world tests? It seems different Manning values are used for the two tests. Please clarify how the optimisation processes is applied for determining these parameters.*

**Response:** The standard approach to modelling floods in urban environments is to calibrate the roughness coefficient to observed parameters of the flood. However, actual values of the friction parameters will be model and possibly scale-dependent as within models of varying complexity, friction values account for a variety of artefacts and unrepresented processes. Therefore for the real world urban flood modelling, the Manning's parameters can be different for different cases.

For the Carlisle test, 183-point measurements of maximum water surface elevation is available, which were applied for the calibration of the Manning's roughness values. A total of 78 simulations was conducted that form a matrix of thirteen effective n values for the channel model (0.01, 0.02, 0.03, 0.035, 0.04, 0.045, 0.05, 0.055, 0.06, 0.065, 0.07, 0.075 and 0.08) and six effective n values for the floodplain model evenly spaced between 0.02 and 0.12. These ranges were deliberately large to ensure the optimum was bracketed. When simulated inundation extent did not reach a wrack or water mark, the water surface elevation of the nearest wet cell was used to calculate the simulation error. This pair of Manning's roughness values for the floodplain and river channels which result in the smallest RMSE compared with the surveyed water marks are selected as the optimised roughness value. The optimised Manning parameter for Carlisle test is $0.04\text{m}^{-1/3}\text{s}$ for the channel and $0.06\text{m}^{-1/3}\text{s}$ for the floodplain. A detailed calibration process for Carlisle test can be found in Neal et al. (2009).

For Glasgow test, a similar calibration process are performed. The flood inundation extent from (Hunter et al., 2008) is applied as the reference, and different pairs of roughness coefficients chosen from a wide but physically plausible range is set up to mimic the effect of typical calibration procedures. The Manning roughness for smooth street varies from 0.01 to 0.03 while other areas varies from 0.01 to 0.08, both in steps of 0.005. This results in the optimised Manning's roughness for the smooth streets is 0.02 while other area has a roughness coefficients of $0.05\text{m}^{-1/3}\text{s}$.

These two calibration processes for the Manning's roughness are now included in the manuscript.

*4, Please clarify why a rectangular river channel is included, instead of other shape like a trapezoidal channel. It is quite common to have a trapezoidal shape in real world. Will the shape of river channel affect the modelling accuracy and why?*

**Response:** In fact we have different shapes of the river channel in the subgrid channel model, for example the rectangular channel, triangular channel and parabolic channel and etc. The different shapes of river channel would slightly impact the flow conveyance capacity, due to the equation difference regarding of the specific shape that used for calculating the flow area, wetted perimeter and hydraulic radius. However, the rectangular river channel is set as default in subgrid channel model and the reasons are as follows:

(1), as we are less concerned about the details of the in-channel flow, the river channel is simply routing the flood volume downstream, and identify the locations where the overbank flow happens. Even with an advanced shape, the river hydraulics are calculated in a simplified 1D approach, and no obvious superiority can be expected with an advanced channel shape instead of a rectangular one.

(2), for the river bathymetry estimation, we always demand an equal area as the surveyed cross sections. The estimated bathymetry should quite close to the observed one, even with different shapes. In practice, we can get the river width and depth much more easily compared with other parameters like the bottom width for a trapezoidal channel. Therefore the rectangular river channel helps to estimate the river bathymetry in a way that are consistent with the observations.

*5, It should be × instead of \* in Equation (10).*

**Response:** Corrected as suggested.

*6, In line 408, lower case of the "A".*

**Response:** Corrected as suggested.

*7, In line 38/353/435, there is a space between the value and its unit, while in other areas no space is added. Please check and make sure uniform format is applied.*

**Response:** Corrected as suggested.

**Reference:**

Hunter, N., Bates, P., Neelz, S., Pender, G., Villanueva, I., Wright, N., Liang, D., Falconer, R. A., Lin, B., and Waller, S.: Benchmarking 2D hydraulic models for urban flooding, Proceedings of the Institution of Civil Engineers-Water Management, 13-30,
Neal, J., Schumann, G., and Bates, P.: A subgrid channel model for simulating river hydraulics and floodplain inundation over large and data sparse areas, Water Resources Research, 48, 2012.
Neal, J. C., Bates, P. D., Fewtrell, T. J., Hunter, N. M., Wilson, M. D., and Horritt, M. S.: Distributed whole city water level measurements from the Carlisle 2005 urban flood event and comparison with hydraulic model simulations, Journal of Hydrology, 368, 42-55, 2009.